# 2-Phenethylamines in Medicinal Chemistry: A Review

**DOI:** 10.3390/molecules28020855

**Published:** 2023-01-14

**Authors:** Carlos T. Nieto, Alejandro Manchado, Leland Belda, David Diez, Narciso M. Garrido

**Affiliations:** Department of Organic Chemistry, Faculty of Chemical Sciences, University of Salamanca, Pl. Caídos, s/n, 37008 Salamanca, Spain

**Keywords:** 2-phenethylamine, medicinal chemistry, ligands, adrenoceptors, carbonyl anhydrase, dopamine receptor, DAT, 5-HT, MAO, PPAR, sigma receptors, TAAR1

## Abstract

A concise review covering updated presence and role of 2-phenethylamines in medicinal chemistry is presented. Open-chain, flexible alicyclic amine derivatives of this motif are enumerated in key therapeutic targets, listing medicinal chemistry hits and appealing screening compounds. Latest reports in discovering new bioactive 2-phenethylamines by research groups are covered too.

## 1. Introduction

The 2-phenethylamine motif is widely present in nature, from simple, open-chain structures to more complex polycyclic molecular arrangements. The importance of this moiety is probably best exemplified by the endogenous catecholamines dopamine, norepinephrine and epinephrine (an example of open-chain 2-phenethylamines), exhibiting a central role in dopaminergic neurons, which play a critical role in voluntary movement, stress or mood [1]. Several naturally occurring alkaloids, i.e., morphine, (*S*)-reticuline or berberine, embedded in the 2-phenethylamine unit form more complex cyclic frameworks derived from its natural biosynthetic pathways (Figure 1).

Additionally, in addition to their prominent therapeutic applications, it is worth mentioning the recreational use of a long list of alkaloids incorporating the aforementioned moiety (“designer drugs”) [2], responsible for drug abuse-related conditions [3,4,5,6,7,8]. Surprisingly, the literature lacks a comprehensive summary that bundles up 2-phenethylamine-based structures and known therapeutic targets, including basic hits or advanced leads. Pairing these will present an appealing opportunity to both new and experienced researchers to summarize 2-phenethylamines target binding and therapeutic scope, as well as selectivity/antitarget issues. Considering all these, a review covering the medicinal chemistry landscape is presented here as a brief, central resource linking up 2-phenethylamine hits and receptors. From the structural point of view, 2-phenethylamines present a vast therapeutic chemical space, not just as is, but considering different substitutions, functional group decorations, ring enclosures or heteroaromatic analogues. Describing such a massive quantity of scaffolds with the phenethylamine resemblance is beyond the scope of this review and more, such asly requires a dedicated book. For this reason, the present review covers only structures where there is an alicyclic amine (Figure 2, blue examples), clustering them all under the 2-phenethylamine label. The authors believe that such cases as 2-heteroaryl ethylamines or cyclic amines, despite presenting 2-phenethylamine resemblance, represent other categorized entities (i.e., tetrahydroisoquinolines or 3-phenyl-pirrolidines examples) on their own, worth independent review (Figure 2, red examples). This review also covers those 2-phenethylamines where the phenyl ring is condensed with a heteroaryl ring, as the primary motif ring is a phenyl one.

This review is divided in sections covering those molecular targets where 2-phenethylamines were found to be biologically relevant. Medicinal chemistry leads and state-of-the-art research on novel molecules are described here.

## 2. 2-Phenethylamine Targets of Biological Importance

### 2.1. Adenosine Receptors

Adenosine receptors family are G-protein-coupled receptors (GPCR) widely distributed in human body tissue. They have four members, named A_1_, A_2A_, A_2B_ and A_3_, with well-reported activities in mediating inflammation, cardiovascular vasodilation or central and peripheral nervous system pathological responses [9,10,11,12].

The 2-phenethylamine moiety may be found in a range of AR (adenosine receptors) ligands, such as *N*6-(2-phenylethyl)adenosine (**1**) [13], APNEA (*N*6-[2-(4-ainophenyl)ethyl]adenosine) (**2**) [14,15], CGS 21680 (**3**) [16,17,18,19] or ZM241385 (**4**) [20,21,22]. Murai et al. synthesized photoreactive CGS 21680 derivatives comprising photophores, such as benzophenone **5** or phenylazide **6** for photoaffinity labeling (Figure 3). This allows elucidation of the functional analysis of adenosine receptor A_2A_ through competitive binding assays against [^3^H]-NECA (*N*-ethyladenosine-5′-uronamide) [23].

### 2.2. α-Adrenergic Receptors

Another class of GPCR targeted by 2-phenethylamines are constituted by α-adrenergic receptors (or α-adrenoceptors). There are two main groups of α-adrenergic receptors, α_1_ and α_2_, with several subtypes within (α_1A_, α_1B_, α_1D_, α_2A_, α_2B_, α_2C_). Group α_1_ is distributed in cardiovascular, intestinal, CNS and urinary systems, while group α_2_ is located in pancreas, CNS, and cardiovascular regions as well [24].

Probably the best representatives of the 2-phenethylamine chemical space are the endogenous catecholamines L-DOPA (**7**), dopamine (**8**), norepinephrine (**9**) (noradrenaline) and epinephrine (**10**) (adrenaline), biosynthetically produced in cascade from phenylalanine (**11**)/tyrosine (**12**) naturally occurring amino acids (Figure 4).

Based on these catecholamines, several studies were performed to investigate the effect of chirality, further functionalization, and activity on different derivatives [25,26,27,28,29,30,31]. This later triggered the elaboration of many derivatives conserving the 2-phenylethyl moiety, which have been frequently used in the context of medicinal chemistry as tool compounds (Figure 5, Table 1).

### 2.3. β-Adrenergic Receptors

Close to the previously described α-type, the β-adrenergic receptors are also activated by catecholamines norepinephrine and epinephrine. There are three receptor subtypes (β_1_, β_2_ and β_3_) that are implicated in diverse cardiovascular and pulmonary functions [53,54], leading to a vast ligand chemical space (Figure 6, Table 2) to treat cardiogenic shock, heart failure, asthma, overactive bladder (agonists), arrhythmias, hypertension (antagonists commonly known as beta blockers) [55].

### 2.4. Aldose Reductase

ALR2 aldose reductase is an enzyme of the polyol pathway responsible for the transformation of glucose into sorbitol, with relevant involvement in long-term diabetic complications. A series of modified 2-phenethylamines **85** comprising the insertion of aliphatic chains, aromatic rings or carboxylic acids were elaborated by Sun et al. [130]. This resulted in the obtention of a small library of derivatives with low inhibition effects towards in vitro pig kidney ALR2 (Figure 7).

### 2.5. Carbonic Anhydrase

Carbonic anhydrases (CAs) are Zn-based (also Fe-based) metalloenzymes present across all living organisms of the different life kingdoms. Divided in eight different families (α, β, γ, δ, ε, ζ, η, θ, and t types), they catalyze the hydration of carbon dioxide to bicarbonate, with the purpose of transporting CO_2_ to HCO_3_^-^ between tissue types, contributing to pH homeostasis, bone calcification and electrolyte transport, and ultimately participate in biogenic routes/processes, such as lipogenesis, ureagenesis or gluconeogenesis [131,132,133].

Several groups have independently used 2-phenethylamine based sulfamides and monothiocarbamates (Figure 8) as CA inhibitors, as frequently targeting a specific CA is related to a certain syndrome or disease, such as glaucoma [134], obesity [135], or certain types of cancer [136].

Nocentini et al. tested phenethylamine monothiocarbamates **87** and **88** as well as other cyclic derivatives against human CA I/II (hCA), with 26–43 nM activities in type II (Figure 8a) [137]. Symmetric sulfamides were employed by Topal et al. (Figure 8b) with hCAI/II inhibition demonstrated at nanomolar level [138]. Branched sulfamides integrating the 1-phenyl-2-phenethylamine scaffold, elaborated by Akıncıoğlu et al. [139] were found to be single-digit nanomolar inhibitors of both type I/II hCA (Figure 8c).

### 2.6. Dopamine β-Hydroxylase

Dopamine β-hdroxylase (DBH) is a Cu-based oxidoreductase that controls dopamine transformation into norepinephrine (Figure 2) in several neuron types (like adrenergic or noradrenergic ones) [140]. Limited studies have been developed on the use of modified 2-phenethylamines to target DBH. Kruse et al. [141] developed 2-vinyl- and 2-alkynyl-based 2-phenethylamines with moderate vitro activities (Figure 9).

More advanced DBH inhibitors are constituted by imidazolethione amines, such as etamicastat [142,143], nepicastat [144] or zamicastat [145], with low resemblance to dopaminergic amines.

### 2.7. Dipeptidyl Peptidases (DPP)

Dipeptidyl peptidases are exopeptidases responsible for proteolytic transformations, specifically cleaving the peptide bond after the penultimate proline residue. DPP4 is a serine protease displaying a critical role in cell adhesion, inflammation processes and immune regulation by deactivating GLP-1 and hence lowering blood glucose levels [146]. Type 2 diabetes is the main therapeutic area were DPP4 ligands have been discovered (Figure 10 and Figure 11, Table 3).

Backes et al. [154] and and Pei et al. [155] developed pyrrolidine-constrained and piperidine-constrained phenethylamines selectively targeting DPP4 with interesting PK profiles (Figure 11).

### 2.8. Dopamine Receptors (DX)

Dopamine receptors are a class of GPCR widely distributed in the brain, with key functionalities related to cognition, motivation and muscular drive. Pharmacologically, they are grouped into two families: D1-type (D1 and D5 receptors) and D2-type (D2S, D2L, D3 and D4) [156,157]. Medicinal chemistry of D1/D2-type receptor ligands addresses mainly the treatment of schizophrenia. From the chemical point of view concerning this review, a few small molecules presenting a basic 2-phenethylamine structure are reported in the literature (Figure 12, Table 4).

### 2.9. Dopamine Transporter (DAT)

The dopamine transporter receptor modulates the availability of released dopamine in the synaptic space by relocating it back into the presynaptic cell. It serves as a main target for recreational drugs as well as psychostimulant and antidepressant drugs. Classically, DAT ligands are classified in amphetamine-type and cocaine-type structures [164]. While cocaine-type molecules exert inhibitory binding to DAT, amphetamine-type ones are substrates that are effectively transported to the presynaptic neuron, stimulating efflux of cytosolic dopamine [165].

Amphetamine-type DAT ligands can be grouped in two families: amphetamine derivatives **107** and cathinone derivatives **114**. Amphetamines are 1-alkyl-2-phenethylamine derivatives **107** summarized below (Figure 13, Table 5). SAR attributes [166] are well known in this series, with diversification of the biological response with aryl substitutions, pharmacokinetic parameter shift upon alkylation of the amino group and decrease in dopaminergic pathways influenced by elongation of the alkyl chain at position 1. Due to being controlled substances in most countries, the number of compounds is constantly increasing, trying to avoid the introduction of compounds in the corresponding prohibition lists of each state [164,167]. Major concerns of this series are neurotoxicity, myocardial infarction, aneurysms, pulmonary hypertension, and tooth decay [168].

### 2.10. Galectin-1 Receptor

Galectins are a family of soluble carbohydrate binding proteins with several roles in inflammation, immune response, autophagy or signaling. Comprising 16 members, only 12 are expressed in humans [171]. Tejler et al. [172] described the synthesis of lactose derivatives, such as **121** with the phenethylamine moiety by 1,3-dipolar cycloadditions, with selective galectin-1 inhibition (Figure 14).

### 2.11. HIV-1 Reverse Transcriptase Receptor

Human immunodeficiency virus (HIV), origin of acquired immunodeficiency syndrome (AIDS), is a single-stranded (ss) RNA virus whose infection and propagation mechanisms requires reverse transcription into double-stranded (ds) DNA as a critical stage [173]. HIV-1 reverse transcriptase receptor (HIV-1 RT) is a well-explored target for antiretroviral therapies [174].

Venkatachalam et al. [175] designed a library of phenethyl thiourea compounds with good potency against HIV-1 RT inhibition without any evidence of cytotoxicity [175,176,177]. Eventually (Figure 15), these derivatives evolve in the phenethylthiazolylthiourea (PETT) family, with trovirdine as its prominent inhibitor [178,179,180].

### 2.12. 5-Hydroxytriptamine (5-HT) Receptors

5-Hydroxytryptamine (5-HT) receptors are one of the most extensively studied receptor families, with seven subtypes (5-HT_1_, 5-HT_2_, 5-HT_3_, 5-HT_4_, 5-HT_5_, 5-HT_6_, 5-HT_7_) identified [181]. Therapeutic indications of 5-HT ligands and advanced leads (Figure 16, Table 6) cover different conditions, such as migraine, depression, social phobia, obsessive–compulsive disorder, anxiety, schizophrenia, eating disorders, panic-disorders, hypertension, pulmonary hypertension, emesis, vomiting, and irritable bowel syndrome (IBS) [182].

A novel class of 2-phenethylamines with hallucinogenic/psychedelic effects are N-benzylphenethylamines or NBOMes (**133**, **134**) (Figure 17) [195]. These agents have a selective binding profile towards 5-HT_2_ receptor subtypes (5-HT_2A_, 5-HT_2B,_ 5-HT_2C_), making them promising therapeutic compounds. Traditionally, the assumption of converting the primary amine into a secondary one was associated with a prominent loss in 5-HT_2A_ activity. N-benzyl substitution was found to be the exception, increasing affinity and potency at the receptor [196]. SAR exploration of the NBOMes scaffold led to defining avoidable regions for SAR expansion, while mapping tolerated substitutions seeking potency/selectivity [197]. From the original 25X-NBOMe halide derivatives [198], different derivatives have evolved.

Jensen et al. developed 25CN-NBOH (**135**) (Figure 17a) as a result of halide substitution by the cyano moiety, displaying high-picomolar/low-nanomolar binding affinities (competition binding assays with [^3^H]ketanserin) and functional potencies at 5-HT_2A_ receptor. Leth-Petersen et al. [199] designed a library of 25B-NBOMe analogues **136**, such as **137** or **138** in the search of decreasing intrinsic clearance (Figure 17b). Despite the authors’ efforts to decrease intrinsic clearance by lipophilicity reduction, no correlation was found, although several 5-HT_2_ potent compounds were synthesized. Nichols et al. [200] explored the impact of methoxy and bromo scanning along the benzyl ring of 25I-NBOMe (**139**). Ortho or meta positions enhanced activity, whereas the para substitution reduced it. One of the best derivatives was **140**, which was compared with its tryptamine congener **141**, less potent overall in the 5-HT_2_ assays performed (Figure 17c).

NBOMes and polyalkoxylated phenethylamines could be envisaged as mescaline (Figure 16, **142**) evolving structures. In this sense, significant efforts have been made to derive rational SAR maps together with attractive bioactive chemical matter [201]. Marcher-Rørsted et al. [202] reported the insertion of 2,5-dimethoxy motif in phenethylamine-like 5-HT_2A_ agonists. They demonstrated that this motif is relevant for in vivo potency, but without observed correlations in affinity or potency in competition binding assays (Figure 18a). Oxygen-to-sulfur exchange reduces hallucinogenic-associated activity [203], while removal of one of the 2- or 5-position methoxy groups decreased in vivo activity [166]. Porter et al. [204] have derived 3-amino-chromanes and tetrahydroquinolines as selective 5-HT_2B_/5-HT_7_ ligands (Figure 18b). 5-HT_2B_ is not considered an optimal target, due to valvular heart disease and myofibroblast proliferation by long-term consumption of selective agonists [205,206]. 5-HT_7_ is implicated in sleep, mood and circadian rhythm functions [207]. Kolaczynska et al. [208] analyzed the impact in 5-HT activity of 4-alkoxy exploration of 2,5-dimethoxyphenethylamines and amphetamines (Figure 18c). Both derivatives were found to interact strongly and selectively with 5-HT_2A_, demonstrating that size and lipophilicity increase in this region favors 5-HT_2A/C_ affinity. Schultz et al. [209] and Nichols et al. [210] explored the fusion of furane/pyrane rings with the aromatic ring, as probes of the binding pocket size of 5-HT_2A_ receptor subtype and lone pair suitable orientation of the 2,5-oxy substituents, furnishing nanomolar-range receptor affinities (Figure 18d). McLean et al. [211] portrayed the conformationally restriction of 2-phenethylamines via 1-aminomethylbenzocycloalkanes syntheses (Figure 18e). Benzocyclobutene derivative **153**’s strong potency against 5-HT_2A_ showed the hypothesis that the side chain of the phenethylamine binds in an out-of-the-plane conformation. Some of these conformationally restricted phenethylamines exhibit affinity for muscarinic receptors [212].

### 2.13. Monoamine Oxidase (MAO) Receptors

Monoamine oxidases (MAO) are flavin-containing enzymes that catalyze the oxidative deamination of monoamines, which are bound to the outer membrane of mitochondria. Common MAO substrates are 5-hydroxy-tryptamine and catecholamines (dopamine, norepinephrine and epinephrine). MAO A and MAO B are the two enzyme isoforms, sharing a 70% sequence identity and differentiating each other in the substrate scope [213]. Theoretical approaches to rationalize isoform selection by substrates have been described [214,215]. As these two oxidases are responsible for neurotransmitter inactivation by oxidation, their dysfunction drives several neurological disorders, hence the therapeutic attractiveness (Figure 19, Table 7).

### 2.14. Opioid Receptors

Opioid receptors are a class of GPCR proteins consisting of three receptor types, mainly µ-, δ-, and κ- types, with a variety of functional roles in the nervous system, such as pain signaling, growth, respiration, and immunological response [223,224]. Opioid ligands modulate neuronal inhibition and ultimately analgesia. Relevant side effects are well known, such as constipation or drug dependence/abuse.

Takahashi et al. [225,226,227,228,229,230] derived a small series of flexible 2-phenethylamines with analgesic activities, with moderate potency effects when compared with pentazocine or morphine (Figure 20a). Manchado et al. [231] developed quick asymmetric routes furnishing this type of derivative. Spetea et al. [232] developed selective diphenethylamine-based tertiary amines as κ-opioid receptors with 100-fold and 1000-fold selectivity difference compared with its congeners (Figure 20b).

### 2.15. Peroxisome Proliferator-Activated (PPAR) Receptors

Peroxisome proliferator-activated (PPAR) receptors are peroxisome receptors and subcellular organelles performing several tasks related to cholesterol and fatty acid metabolism. Three members, named PPAR-α, PPAR-δ, and PPAR-γ, form this family, with different levels of expression and functionalities in diverse tissues, from energy storage in endothelial and vascular smooth muscle cells (type γ) to energy expenditure across all bodies (type δ). Several agents have been developed in relation to PPAR to address obesity, inflammation or neurodegenerative disorders (Figure 21, Table 8) [233,234].

### 2.16. Sigma Receptors

Initially described as opioid receptors, sigma receptors conform their own family, unrelated to other receptors. Both members σ1R and σ2R are primarily found at the endoplasmic reticulum, and participate in diverse conditions, such as cancer, pain, neurodegenerative diseases or depression [243]. BD-1047 (Figure 22) is an open-chain, flexible 2-phenethylamine acting as antagonist of σ1R [244].

### 2.17. Trace Amine-Associated Receptors (TAAR)

Relatively recently discovered, trace amine-associated receptors (TAAR) are a GPCR family composed of nine members (TAAR1 to 9) with prospective therapeutic applications in the field of schizophrenia and metabolic disorders [245]. Lewin et al. [246] carried out SAR explorations with simple 2-phenethylamines to envisage improved pharmacological hits (Figure 23).

## 3. Methods

All described compounds, targets and activities were retrieved using “2-phenethylamine” as title or keyword term in the chemical databases SciFinder [247] and Scopus [248]. Additionally, a SciFinder and Scopus structure search, with the scope described early in this review (Figure 2), was employed.

## 4. Conclusions

This review represents a concise, central summary of relevant 2-phenethylamine-based leads and research hits, which spans receptors and their corresponding therapeutic indications. This report serves as a guide to researchers interested in medicinal chemistry to identify suitable ligand–target associations displaying the aforementioned motif, as well as help to identify prospective targets of brand-new molecules with the 2-phenethylamine core embedded.

Future directions will include both a complementary report covering synthetic strategies to access 2-phenethylamine derivatives and a second, satellite review of 2-heteroaryl-ethylamines in medicinal chemistry.

## Figures and Tables

**Figure 1 molecules-28-00855-f001:**
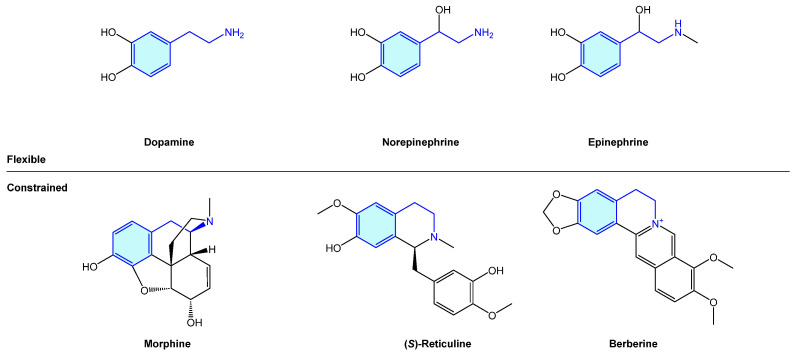
Examples of naturally occurring biologically active compounds displaying a 2-phenethylamine scaffold.

**Figure 2 molecules-28-00855-f002:**
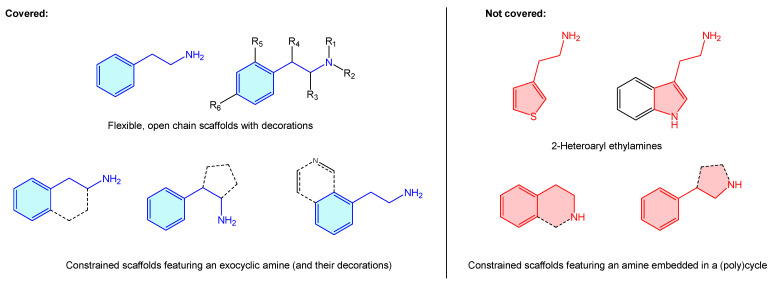
Description of 2-phenethylamine scope of the present review.

**Figure 3 molecules-28-00855-f003:**
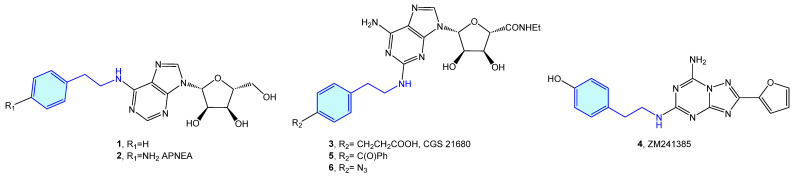
2-Phenylethyl-based AR ligands.

**Figure 4 molecules-28-00855-f004:**
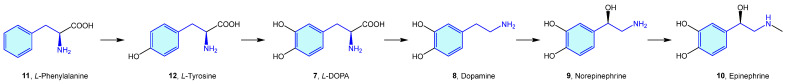
Catecholamine biosynthetic pathways.

**Figure 5 molecules-28-00855-f005:**
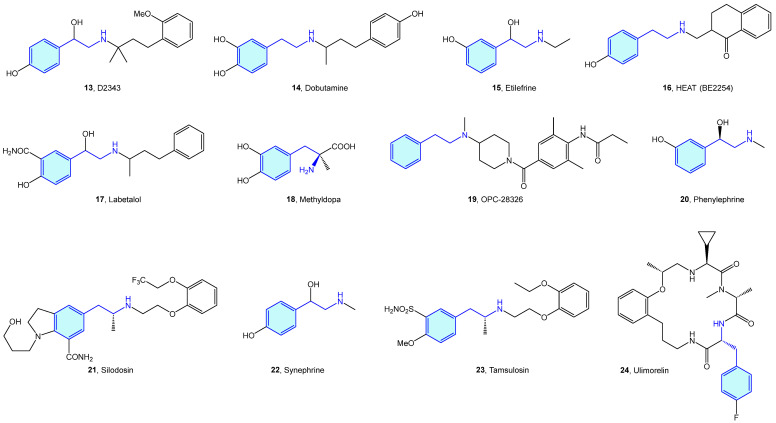
2-Phenylethyl-based α-adrenergic medicinal chemistry leads.

**Figure 6 molecules-28-00855-f006:**
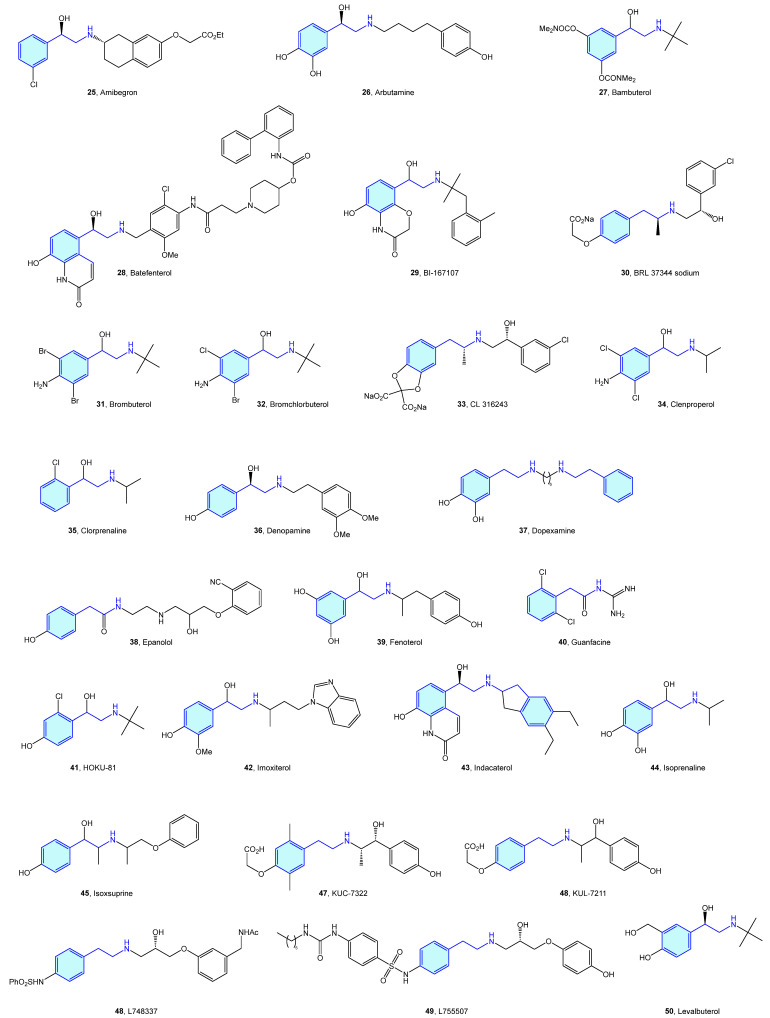
2-Phenylethyl-based β-adrenergic medicinal chemistry leads.

**Figure 7 molecules-28-00855-f007:**
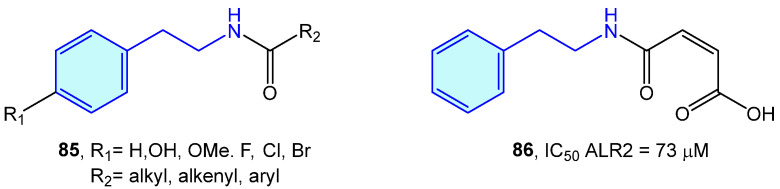
2-Phenylethyl-based ALR2 ligands by Sun et al [130].

**Figure 8 molecules-28-00855-f008:**
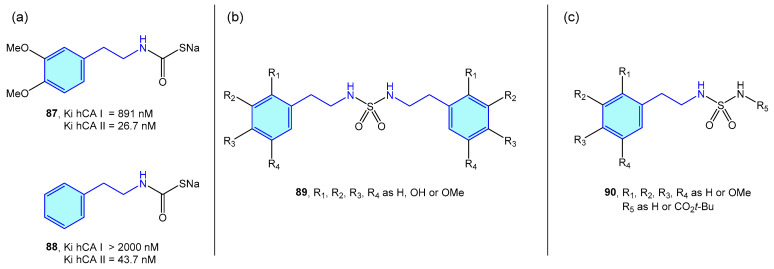
2-Phenylethyl-based CA ligands.

**Figure 9 molecules-28-00855-f009:**
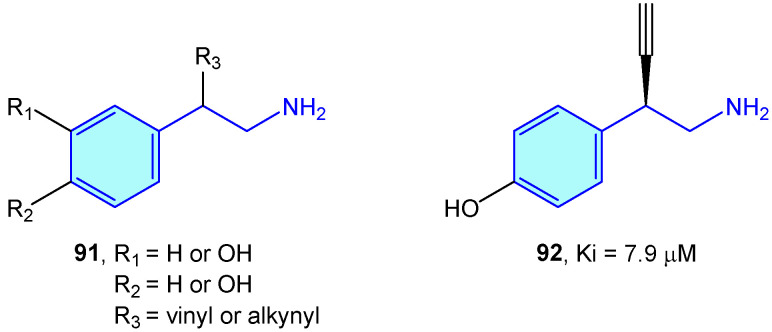
2-Phenylethyl-based DBH ligands by Kruse et al [141].

**Figure 10 molecules-28-00855-f010:**
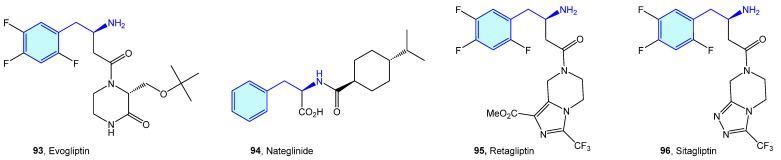
2-Phenylethyl-based DPP4 medicinal chemistry leads.

**Figure 11 molecules-28-00855-f011:**
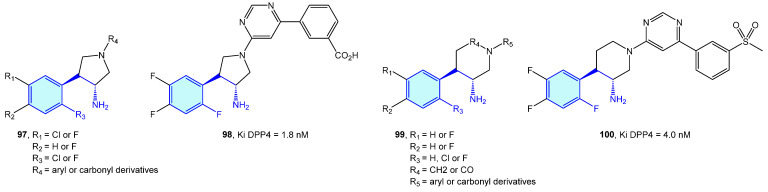
2-Phenylethyl-based DPP4 receptor ligands.

**Figure 12 molecules-28-00855-f012:**
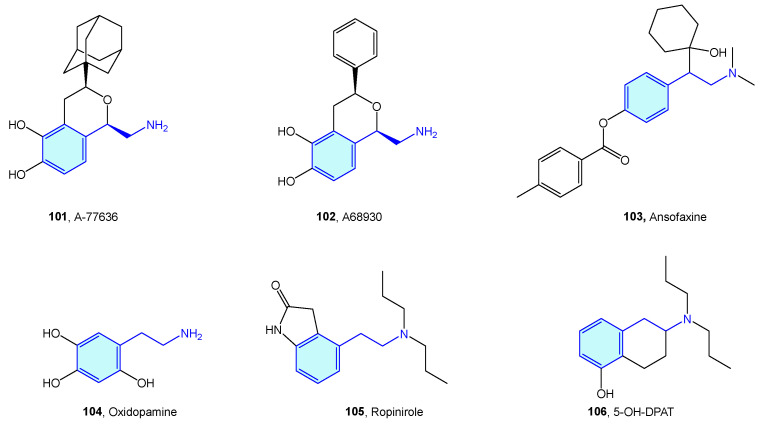
2-Phenylethyl-based dopamine receptor medicinal chemistry leads.

**Figure 13 molecules-28-00855-f013:**
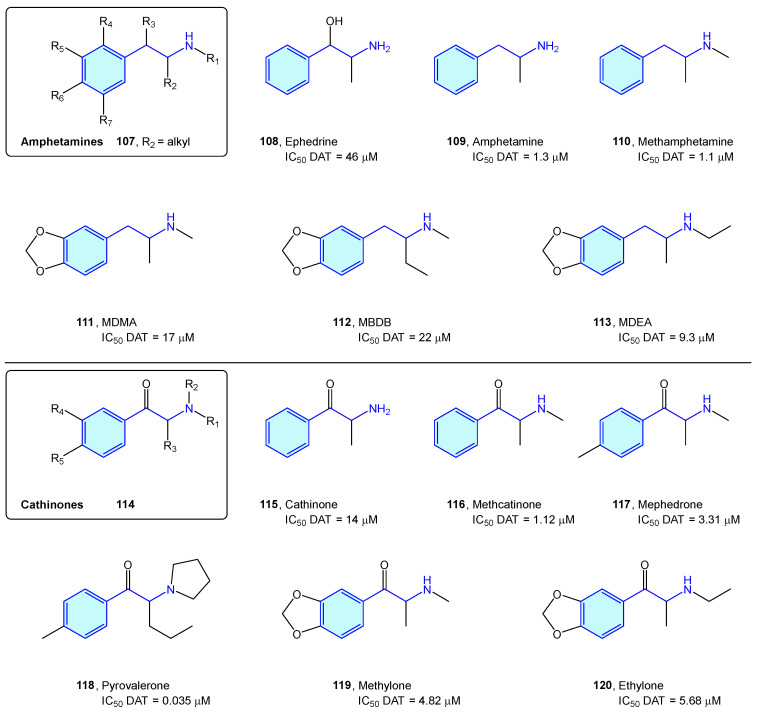
Amphetamine/cathinone-type families.

**Figure 14 molecules-28-00855-f014:**
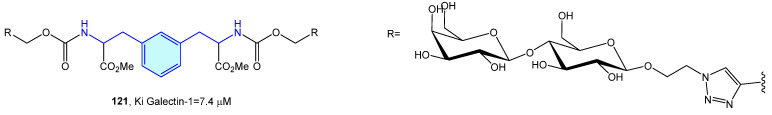
2-Phenylethyl-based galectin-1 receptor ligands.

**Figure 15 molecules-28-00855-f015:**
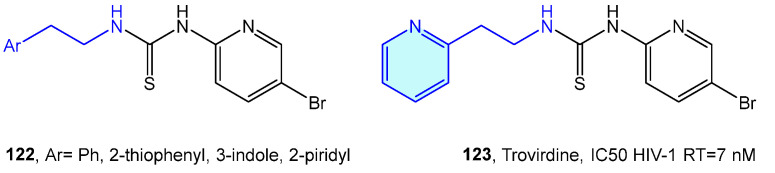
2-Phenylethyl-based HIV-1 RT receptor ligands.

**Figure 16 molecules-28-00855-f016:**
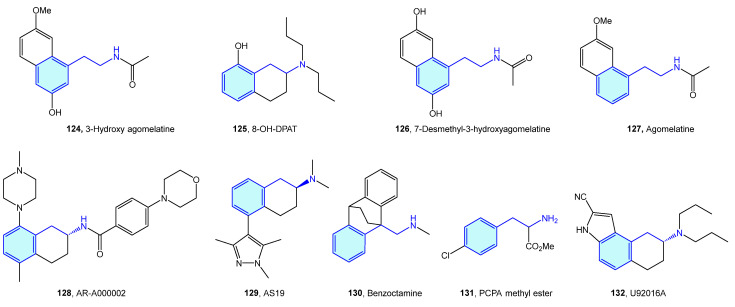
2-Phenylethyl-based 5-HT medicinal chemistry leads.

**Figure 17 molecules-28-00855-f017:**
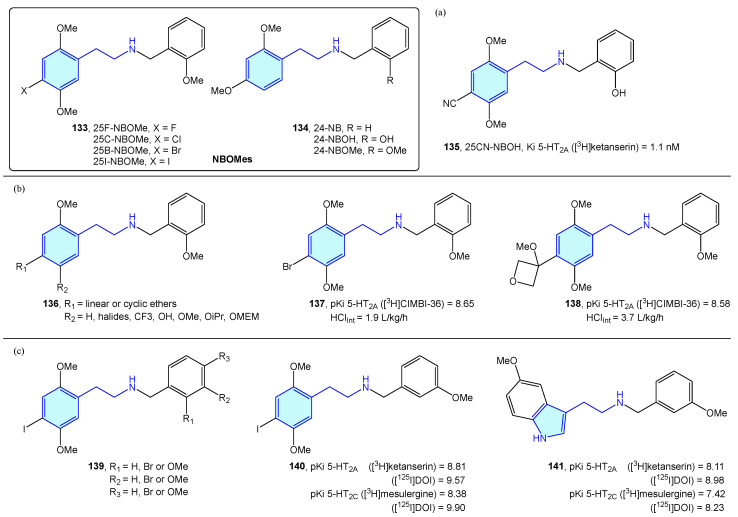
NBOMes with 5-HT receptor activity.

**Figure 18 molecules-28-00855-f018:**
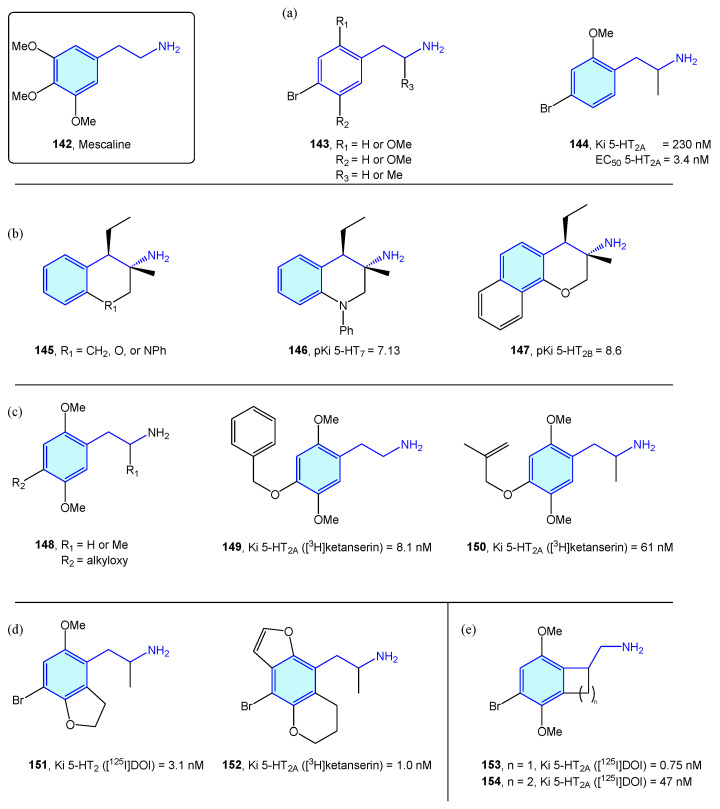
2-Phenethylamines with 5-HT receptor activity.

**Figure 19 molecules-28-00855-f019:**
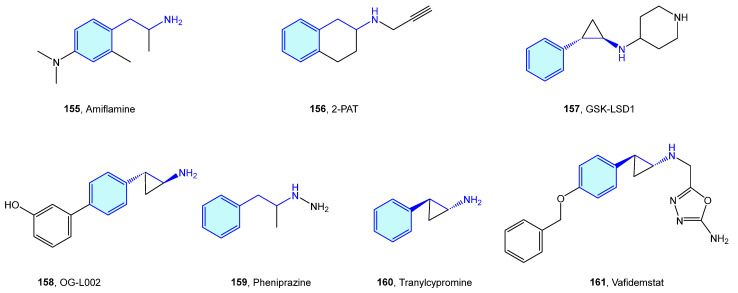
2-Phenethylamine MAO medicinal chemistry leads.

**Figure 20 molecules-28-00855-f020:**
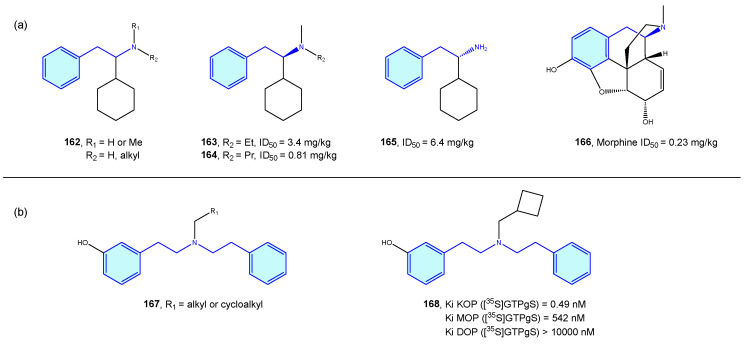
2-Phenethylamines with 5-HT receptor activity.

**Figure 21 molecules-28-00855-f021:**
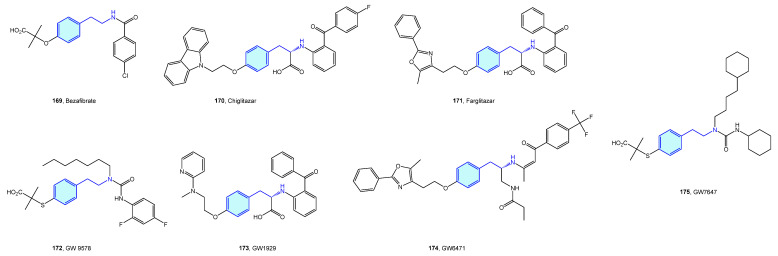
2-Phenethylamine PPAR medicinal chemistry leads.

**Figure 22 molecules-28-00855-f022:**
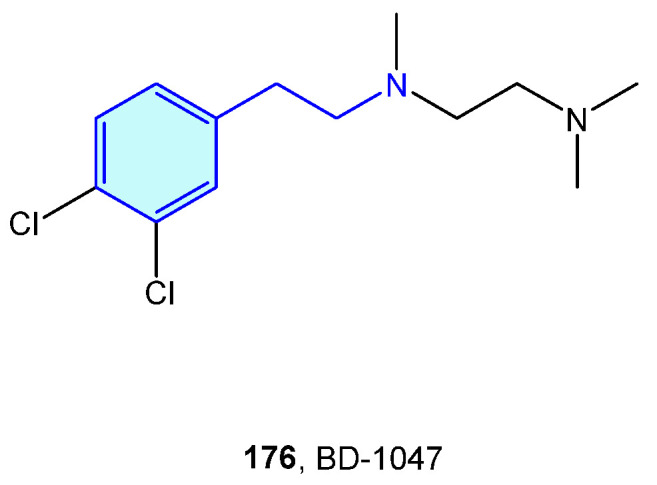
A 2-Phenethylamine Sigma-1 receptor hit.

**Figure 23 molecules-28-00855-f023:**
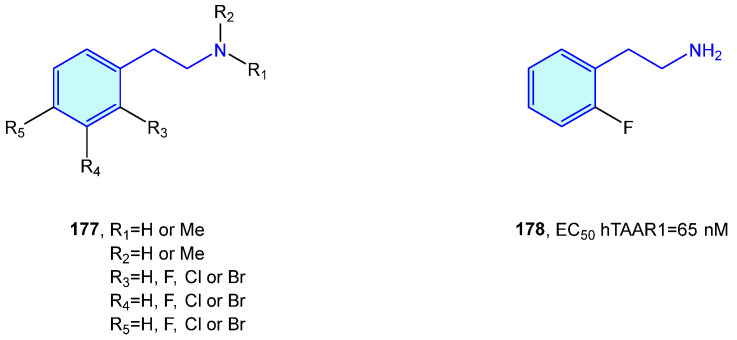
2-Phenethylamines with hTAAR1 bioactivity.

**Table 1 molecules-28-00855-t001:** α-Adrenergic medicinal chemistry leads.

Molecule	Name	Primary Targets	Secondary Targets	References
**13**	D2343	α_1_	β_2_-adrenoceptor	[32]
**14**	Dobutamine	α_1_	β_1_, β_2_-adrenoceptors	[33,34,35]
**15**	Etilefrine	α_1_	AMP-activated protein kinase (AMPK)	[36]
**16**	HEAT (BE2254)	α_1_		[37,38,39]
**17**	Labetalol	α_1_	β-adrenoceptors	[40]
**18**	Methyldopa	α_2_		[41,42]
**19**	OPC-28326	α_2_		[43,44]
**20**	Phenylephrine	α_1_		[45,46]
**21**	Silodosin	α_1_		[47,48]
**22**	Synephrine	α_1_	β-adrenoceptors	[49,50]
**23**	Tamsulosin	α_1_		[51]
**24**	Ulimorelin	α_1_	Ghrelin receptor (GRLN)	[52]

**Table 2 molecules-28-00855-t002:** β-Adrenergic medicinal chemistry leads.

Molecule	Name	Primary Targets	Secondary Targets	References
**25**	Amibegron	β_3_		[56,57]
**26**	Arbutamine	β		[58,59,60]
**27**	Bambuterol	β		[61]
**28**	Batefenterol	β_2_	Muscarinic M2, M3	[62,63]
**29**	BI-167107	β_2_		[64]
**30**	BRL 37344 sodium	β_3_		[65]
**31**	Brombuterol	β		[66]
**32**	Bromchlorbuterol	β		[67]
**33**	CL 316243	β_3_		[68,69]
**34**	Clenproperol	β_2_		[70]
**35**	Clorprenaline	β_2_		[71]
**36**	Denopamine	β_1_		[72]
**37**	Dopexamine	β_2_		[73]
**38**	Epanolol	β		[74]
**39**	Fenoterol	β_2_		[75,76]
**40**	Guanfacine	β_1_	α_2_-adrenoceptors	[77]
**41**	HOKU-81	β_2_		[78]
**42**	Imoxiterol	β		[79]
**43**	Indacaterol	β		[80]
**44**	Isoprenaline	β		[81,82,83]
**45**	Isoxsuprine	β	N-methyl-D-aspartate (NMDA)	[84]
**46**	KUC-7322	β_3_		[85]
**47**	KUL-7211	β		[86]
**48**	L748337	β_3_		[87]
**49**	L755507	β_3_		[88]
**50**	Levalbuterol	β_2_		[89]
**51**	Lubabegron	β		[90]
**52**	LY377604	β_3_		[91]
**53**	Mapenterol	β_2_		[92]
**54**	Metaproterenol	β_2_		[93,94]
**55**	Mirabegron	β_3_		[95]
**56**	N-5984	β_3_		[96]
**57**	Naminterol	β_2_		[97]
**58**	Navafenterol	β_2_	Muscarinic M3	[98]
**59**	Octopamine	β		[99,100]
**60**	Olodaterol	β_2_		[101,102]
**61**	Pamatolol	β		[103]
**62**	PF-610355	β_2_		[104,105]
**63**	Phenylethanolamine A	β		[106]
**64**	Pronethalol	β		[107]
**65**	Reproterol	β_2_	Phosphodiesterase (PDE)	[108]
**66**	Ritodrine	β_2_		[109]
**67**	Ro 363	β_1_		[110]
**68**	Rotigotine	β	α_2_-adrenoceptor, 5-HT_1A_, Dopamine D2, D3, D4, D5	[111]
**69**	Salbutamol	β_2_		[112]
**70**	Salmeterol	β_2_		[113]
**71**	SB-206606	β_3_		[114]
**72**	Sibenadet	β	Dopamine D2	[115]
**73**	Solabegron	β_3_		[116]
**74**	Sulfinalol	β		[117]
**75**	Synephrine	β	α-adrenoceptor	[118]
**76**	Talibegron	β_3_		[119]
**77**	TD-5471	β_2_		[120]
**78**	Terbutaline	β_2_		[121,122]
**79**	Tulobuterol	β_2_		[123]
**80**	Vilanterol	β		[124,125]
**81**	Zinterol	β_2_		[126]
**82**	ZK-90055	β_2_		[127]
**83**	-	β_3_		[128]
**84**	-	β_3_		[129]

**Table 3 molecules-28-00855-t003:** DPP4 medicinal chemistry leads.

Molecule	Name	Primary Targets	Secondary Targets	References
**93**	Evogliptin	DPP4		[147,148,149]
**94**	Nateglinide	DPP4		[150]
**95**	Retagliptin	DPP4		[151]
**96**	Sitagliptin	DPP4		[152,153]

**Table 4 molecules-28-00855-t004:** Dopamine receptor medicinal chemistry leads.

Molecule	Name	Primary Targets	Secondary Targets	References
**101**	A-77636	D1		[158]
**102**	A68930	D1		[159]
**103**	Ansofaxine	D		[160]
**104**	Oxidopamine	D		[161]
**105**	Ropinirole	D2, D3, D4		[162]
**106**	5-OH-DPAT	D		[163]

**Table 5 molecules-28-00855-t005:** Dopamine transporter classical binders.

Molecule	Name	References
**108**	Ephedrine	[169]
**109**	Amphetamine	[170]
**110**	Methamphetamine	[170]
**111**	MDMA	[170]
**112**	MBDB	[170]
**113**	MDEA	[170]
**115**	Cathinone	[170]
**116**	Methcathinone	[170]
**117**	Mephedrone	[170]
**118**	Pyrovalerone	[170]
**119**	Methylone	[170]
**120**	Ethylone	[170]

**Table 6 molecules-28-00855-t006:** 5-HT medicinal chemistry leads.

Molecule	Name	Primary Targets	Secondary Targets	References
**124**	3-Hydroxy agomelatine	5-HT_2C_		[183]
**125**	8-OH-DPAT	5-HT_1A_		[184,185]
**126**	7-Desmethyl-3-hydroxyagomelatine	5-HT_2C_	Melatonin MT1, MT2	[186]
**127**	Agomelatine	5-HT	Melatonin MT1, MT2	[187]
**128**	AR-A000002	5-HT_1B_		[188]
**129**	AS19	5-HT_7_		[189,190]
**130**	Benzoctamine	5-HT		[191]
**131**	PCPA methyl ester	5-HT		[192]
**132**	U92016A	5-HT_1A_		[193,194]

**Table 7 molecules-28-00855-t007:** MAO medicinal chemistry leads.

Molecule	Name	Primary Targets	Secondary Targets	References
**155**	Amiflamine	MAO A		[216]
**156**	2-PAT	MAO A, B		[217]
**157**	GSK-LSD1	MAO	Lysine specific demethylase 1 (LSD1)	[218]
**158**	OG-L002	MAO A, B		[219]
**159**	Pheniprazine	MAO		[220]
**160**	Tranylcypromine	MAO		[221]
**161**	Vafidemstat	MAO B	Lysine specific demethylase 1 (LSD1)	[222]

**Table 8 molecules-28-00855-t008:** PPAR medicinal chemistry leads.

Molecule	Name	Primary Targets	Secondary Targets	References
**169**	Bezafibrate	PPAR		[235,236]
**170**	Chiglitazar	PPAR		[237]
**171**	Farglitazar	PPAR-γ		[238]
**172**	GW 9578	PPAR-α		[239]
**173**	GW1929	PPAR-γ		[240]
**174**	GW6471	PPAR-α		[241]
**175**	GW7647	PPAR-α		[242]

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
