# Peer review of "2-Phenethylamines in Medicinal Chemistry: A Review"

_molecules, 2023, doi:10.3390/molecules28020855_

Round 1
Reviewer 1 Report
Summary of the key contribution of the Review:
2-Phenethylamines in Medicinal Chemistry: A Review.in this review covering updated presence and role of 2-phenethylamines in medicinal chemistry is drafted. Open chain, flexible alicyclic amine derivatives of this motif are enumerated in key therapeutic targets, listing medicinal chemistry hits and appealing screening compounds. Latest reports in discovering new bioactive 2-phenethylamines by research groups are covered too.
Highlights:
· The review clearly explains 2-Phenethylamine targets of biological importance
· Figures are well referenced and clear
· In this review catecholamines, several studies were performed to investigate the effect of chirality, further functionalization, and activity on different derivatives.
· This review provides an overview of the later triggered the elaboration of many derivatives conserving the 2-phenylethyl moiety, which have been frequently used in the context of Medicinal Chemistry as tool compounds.
· This review covers the chemical point of view concerning this review, a few small molecules presenting a basic 2-phenethylamine structure are reported in literature respectively.
· The results of this study will facilitate the mechanism of the reactions and identification of key intermediates are discussed, as are developments in reaction conditions.
· Lowlights:
The paper is explained very well, I would say there are no lowlights.
Author Response
We appreciate the referee's comments regarding this review.
Reviewer 2 Report
2-Phenethylamines in medicinal chemistry: A review by Nieto et al. summarises the presence of the motif in various lead compounds, targeting the related receptors. The manuscript includes comprehensive information on the topic. However, the manuscript needs some amendments, including the following.
(1) What are the aims of the review? In L33-34, it says that describing a large quantity of the motif present in many examples will surpass the purpose of the review, however, this doesn’t capture readers’ attention why this review would be useful and important. The authors should describe elaborately their aims.
(2) The manuscript provides coverage, but it should also describe Methods.
(3) The manuscript did not provide implications/future directions for the topic. The authors should describe the significance of their literature analysis.
Minor corrections
L55, specify what AR stands for or do not use it.
L88, remove ‘and heart failure’.
Figure 10, make the aromatic ring in retagliptin consistent in color with other leads.
Figure 22 has not been cited in the text.
Author Response
We thank the referee 2 for the errors appreciated and the very interesting suggestions that he has proposed.
Referee 2: (1) What are the aims of the review? In L33-34, it says that describing a large quantity of the motif present in many examples will surpass the purpose of the review, however, this doesn’t capture readers’ attention why this review would be useful and important. The authors should describe elaborately their aims.
Author´s answer: We included this concise description of motivations and aims of the review, in line with Reviewers' 2 viewpoint.
Surprisingly, literature lacks a comprehensive summary that bundles up 2-phenethylamine-based structures and known therapeutic targets, including basic hits or advanced leads. Pairing these will present an appealing opportunity to both novel and experience researchers to summarize 2-phenethylamines target binding and therapeutic scope as well as selectivity/anti-target issues. Considering all these, a re-view covering the medicinal chemistry landscape is presented here as brief, central resource linking up 2-phenethylamine hits and receptors.
Reviewer 2: L55, specify what AR stands for or do not use it.
Author´s answer: We included the following term (Adenosine receptors)
The 2-Phenethylamine moiety may be found in a range of AR (Adenosine receptors) ligands …
Reviewer 2: L88, remove ‘and heart failure’.
Author´s answer: These words were removed
Reviewer 2: Figure 10, make the aromatic ring in retagliptin consistent in color with other leads.
Author´s answer: We coloured the aromatic ring in line with this commentary
Reviewer 2: Figure 22 has not been cited in the text.
Author´s answer: We addressed this issue by citing the mentioned figure in the main-body.
… carried out SAR explorations with simple 2-phenethylamines to envisage improved pharmacological hits (Figure 22).
Reviewer 2: (2) The manuscript provides coverage, but it should also describe Methods.
Author´s answer: We included a brief description of the search methods used for this work
- Methods
All described compounds, targets and activities were retrieved introducing “2-phenethylamine” as title or keyword term in chemical databases as SciFinder [1] and Scopus [2]. Additionally, SciFinder & Scopus structure search, accordingly with the scope described early in this review (Figure 2), was employed.
Reviewer 2: (3) The manuscript did not provide implications/future directions for the topic. The authors should describe the significance of their literature analysis.
Author´s answer: We added an ending section including a concise resume of key aspects and future directions for the authors.
- Conclusions
This review represents a concise, central summary of relevant 2-phenethylamine-based leads and research hits, which spans receptors and their corresponding therapeutic indications. This report serves as guide to researchers interested in Medicinal Chemistry in identify suitable ligand-target associations displaying the aforementioned motif, as well as help to identify prospective targets of brand-new molecules with the 2-phenethylamine core embedded.
Future directions will include both a complementary report covering synthetic strategies to access 2-phenethylamine derivatives and a second, satellite review of 2-heteroaryl-ethylamines in medicinal chemistry.